# The Soil heat flux sensor functioning checks, imbalances' origins, and forgotten energies.

Bartosz M. Zawilski

CESBIO Université de Toulouse, CNES, CNRS, INRA, IRD, UPS.

Toulouse, 31000, France

*Correspondence to*: Bartosz M. Zawilski (bartosz.zawilski@cnrs.fr)

**Abstract.** Soil heat flux is an important component of the Surface Energy Balance (SEB) equation. Measuring it

requires an indirect measurement. Every used technique may present some possible errors tied with the utilized specific technique, soil inhomogeneities, or physical phenomena such as latent heat conversion beneath the plates especially in desiccation cracking soil or vertisol. The installation place may also induce imbalances. Finally, some errors resulting from the physical sensor presence, vegetation presence or soil inhomogeneities may occur and are not avoidable. For all these reasons it is important to check the validity of the measurements. One quick and easy

way is to integrate results during one year. By consideration of the inert core internal energy conservation law, it is shown that the corresponding integration should be close to zero after a necessary geothermal heat efflux subtraction. However, below plate evaporation and vegetation absorbed water or rainfall water the infiltration may also contribute to the observed short scale or/and long scale imbalance generating convective heat fluxes not sensed by the heat flux sensors. Another energy source is usually not included in the SEB equation: rainfall or irrigation.

Yet its importance for short- and long-term integration is notable. As an example, the most used sensor: Soil Heat Flux Plates (SHFP), is given.

## 1 Introduction

On the surface of the soil, daytime solar radiation and nighttime soil infrared radiation generate an important heat

flux called *G*. This flux is either positive, heat flux going down to the depths of the soil and mainly due to solar heating, or negative, the soil surface temperature drops and therefore a heat flux rises from the ground to the

surface mainly lasting at night. This heat exchange is important as the energy stored in the soil may be used for water evaporation (Penman, 1948, Monteith, 1965). Many processes, especially biological processes such as roots and microbial activities, are temperature-dependent which is directly related to $G$. Also, the knowledge about $G$ is

necessary to check the well-known Surface Energy Balance or Budget (SEB) (Lettau and Davidson, 1957, Lemon, 1963) given by equation 1:

$$R_n - G = H + L_e$$

(1)

With $R_n$ being the net radiation, $H$ is the sensible heat flux into the atmosphere and $L_e$ is the latent heat flow (evaporation).

For the sake of SEB closing, this equation may be completed including the vegetation heat storage $S_C$ and photosynthesis activity $S_P$ (Meyers and Hollinger, 2004). SEB closure allows us to have a quick quality check on all the concerned measurements (Oncley et al., 2002, Oncley et al., 2007).

Depending on the concerned surface and period, all over the different energy fluxes, $G$ part is significant and may reach up to 50% of $R_n$ (Monteith, 1958, Idso et al., 1975, Choudhury et al., 1987). The soil heat flux is not a direct measurement and is not evident as it cannot be done on the surface but, more or less, deeply buried into the soil. Different techniques are employed: flux plates (heat flux sensing thermopiles), calorimetric (temperature temporal variation), temperature gradient or combination (simultaneous calorimetric and gradient measurement or flux plate

and above storage measurement), see Sauer and Horton (2005), for a recent review see Gao et al. (2017). All the used techniques are sensing only *conduction* heat transfer. *Convection* heat transfer is not sensed. The radiation concerns the soil surface and is sensed by a net radiometer and included in $R_n$ and the convection concerns fluids (liquids or gases) and may potentially bias the measurements but usually are not sensed nor included in SEB or $G$ corrections. Appendix B provides a simple example explaining the importance of the convective heat flux

importance.

One of the most used $G$ sensors is the SHFP buried in the soil. As with every sensor, these plates are subject to biases and errors. Some of these errors are specific to the used heat flux plate measurements technology (thermopile), others are rather specific to the surface exchanges and soil inhomogeneities. Whatever the sensor used for $G$ determination, it is important to check if the acquired measurements were representative of the surface

energy exchanges or possibly biased by inhomogeneities. Further considerations deal with the flux plates sensors example.

SHFP sensing temperature differences across their thickness. This temperature difference is proportional to the *conductive* heat flux going through the plate and inversely proportional to the plate's thermal conductance. Nevertheless, because the soil thermal conductivity is not the same as SHFP thermal conductivity (and then its thermal conductance) the heat flux density is deformed and the measurement is biased (Philip, 1961; Sauer et al., 2003). As the soil thermal conductivity changes greatly with soil water content and soil density (Sepaskhah and Boersma, 1979), flux plates have to be periodically calibrated. Nowadays, the commercial self-calibrating SHFP are available and are calibrated by heating their upper side with a deposited thin resistor and then checking the part of the sensed heat versus the part of the produced heat forming a real-time calibration factor. Liebethal (2006) checks the correct functioning of this calibration. However, SHFPs are punctual (only a small surface is sensed), invasives, and subject to bias measurements (Sauer and Horton, 2005). As for every punctual sensor, there should be enough installed plates to ensure a spatially representative measurement. The SHFP's measurement buried at some depth needs to be completed by adding the upper soil layer heat storage to obtain surface soil heat flux (Ochsner et al., 2007). And finally, as the soil heat plates are sensing only sensible heat fluxes by conduction, any evaporation taking place under the plate, water vapor flowing through the soil into the atmosphere is not sensed causing an imbalance of up to 100W/m² (Buchan, 1989, Mayocchi and Bristow, 1995).

Nevertheless, the flux plate placement remains controversial. On the one hand, to avoid sensible heat to latent heat conversion (evaporation or condensation) beneath the plate biasing measurement, numerous authors and adopted the ICOS protocol (Op de Beeck et Al., 2018) are suggesting 5 cm depth burring. On the other hand, Gentine et al. (2012) is indicating a systemic error due to high-frequency solar radiation variation not sensed by deeply buried SHFP or temperature profile sensors and suggest then 2mm depth.

This short note deals with how to assess the correctness of SHFP functioning and highlighted possible imbalances. It does not deal with the soil layer heat storage above the plate which should be measured and added. Other energies than solar radiation energy should be added to the surface energy balance equations if applicable.

**2 Materials and Methods**

Soil heat plates used for these studies were HFP01SC self-calibrating flux plates from Hukseflux Thermal Sensors B.V., Delftechpark 31, 2628 XJ Delft, The Netherlands. The used datalogger was a CR1000 from Campbell

Scientific, Logan, Utah, USA. Autocalibration is triggered every seven hours: for four minutes heating with 1.4 W power.

For comparison of different operational modes, including or not including the data acquired during and immediately after all calibration periods, data are collected by the logger either every one minute and stocked with a flag corresponding to the calibration initialized every seven hours or averaged every 30 minutes including the

calibration periods. This allows checking the influence of the calibration heater inclusion in the collected data. Plates are used on an ICOS cropland site FR-Lam (43°29'47.21"N, 1°14'16.36"E, silty-clay: 50.3% clay, mainly Kaolinite, 35.8% silt, 11.2% sand, 2.8% organic matter according to the classification described by Malterre and Alabert, 1963). Results reported in this paper concern the year 2020 with winter wheat (Triticum aestivum) culture.

### 3 Results and discussion

**3.1 SHFP a posteriori checks.**

Using the SHFP is probably the easiest way for monitoring $G$ and this point may explain the relative popularity of this technique. In this paper, only the soil flux plate functioning is described and no consideration is given to the above soil heat storage measurement which is another challenge.

In ideal conditions, the soil temperature changes seasonally but after one year it recovers its initial temperature

whatever is the sensed soil temperature-depth. Of course, it is an approximation because there are no two identical years and the soil temperature may vary slightly from one year to another. By simplification, if we are assuming the heat stored in the soil does not change after one year, then the total sensed surface heat flux exchange should be negative due to the geothermal heat flux as explained in the next paragraph and in the appendix A where an explanation of an annual heat exchanges integration nullity is provided. This point is crucial for the later

assessment of a non-sensed or biased heat fluxes measurement. For the rest of this paper, by convention, for the long-term important heat fluxes correction, a superscript "L" is added and for short-term important heat flux corrections, a superscript "S" is added. When a correction is important for both, short- and long-term measurements, no superscript annotation is added.

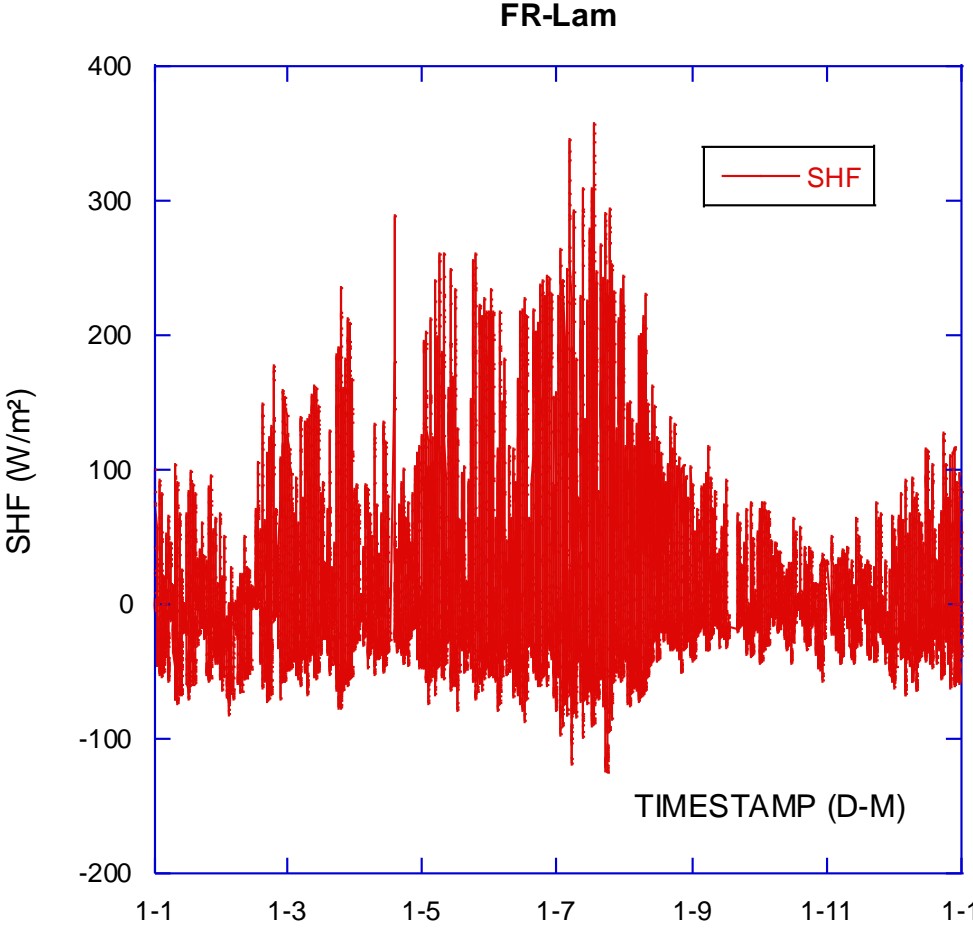

 **Figure 1. Soil Heat flux measured by a self-calibrated heat flux plate for one year.**

**3.2 Heat flux origins and imbalances**.

Indeed, SHFP sensed soil heat flux is not nil since it includes the geothermal heat flux $G_{TH}^{L}$ emitted by the Earth (Elder, 1965). On average, the soil emits 82 mW/m² which is -25 MJ/m² a year depending on the geolocalization.

Figure 1 depicts the soil heat flux recorded by one of our SHFPs installed at the border of an enclosure and considered as a "reference" for data gap filling when other plates have to be temporarily removed (soil operation on cropland). It is difficult or even impossible to know if the measurements are valid based only on that figure. As described in Appendix A once the geothermal contribution is subtracted, the annual integration of a one-dimensional soil heat flux sense on the soil surface should be nil (please see appendix A for more details). Using an integration of the concerned measures, after $G_{TH}^L$ subtraction, as $G_{TH}^L$ is negative we can write:

$$G^C = G - G_{TH}^L = G + |G_{TH}^L|$$

(2)

During one year, starting from zero, we should also end the year at zero (Fig. 2). The geothermal heat flux varies strongly on the Earth's surface being localization specific. In our case it is about -75 mW/m² ($W$ = -24 MJ/m² a year). Section 3.2.5 shows the geothermal correction on FR-Lam which is not negligible even if the geothermal heat flux is relatively small. As we can see in Fig. 2, SHFP, geothermally corrected, $G^C$ measurements integration is not nil and the geothermal energy correction make the imbalance even worse. Far to be negligible, the observed imbalance represents about 10% of the integrated absolute sensed soil heat flux.

The same plate emplacement gives an imbalance more or less important during different years but still always largely positive and represents always about 10% of the integrated absolute flux. The observed largely positive imbalance may be tied to the heat flux plate technique and the installation emplacement. Indeed, Ochsner et al. (2006) compared different methods and reported the main error sources for SHFP; thermal conductivity causing a possible heat flux distortion, a thermal contact between the plate and the soil, latent heat loss, and water (liquid or vapor) flow disruption. Both, the difference between surrounding soil plate thermal conductivity and the poor thermal contact can be overcome by self-calibrating plates. Theoretically speaking, the geothermally corrected overall soil heat flux $G^C$ annual integration should be nil and the possible imbalance has two distinct origins.

- The presence of horizontal heat fluxes resulting mainly from a narrow soil or energy apport inhomogeneity, such as a partially shadowed surface, are described in section 3.2.1. The sensed imbalance is real but the measurement is not valid as the heat flux is no more perpendicular to the plate surface (no more vertical). The overall measurement should include plates on both sides of the inhomogeneity to accurately represent soil heat flux.

-    The convective, not sensed, heat fluxes such as beneath plate evaporation, root pumped water, rainfall water infiltration, and so on, are described in the sections from 3.2.2 to 3.2.4. The corresponding measurements leak should be assessed and added for sake of SEB closure.

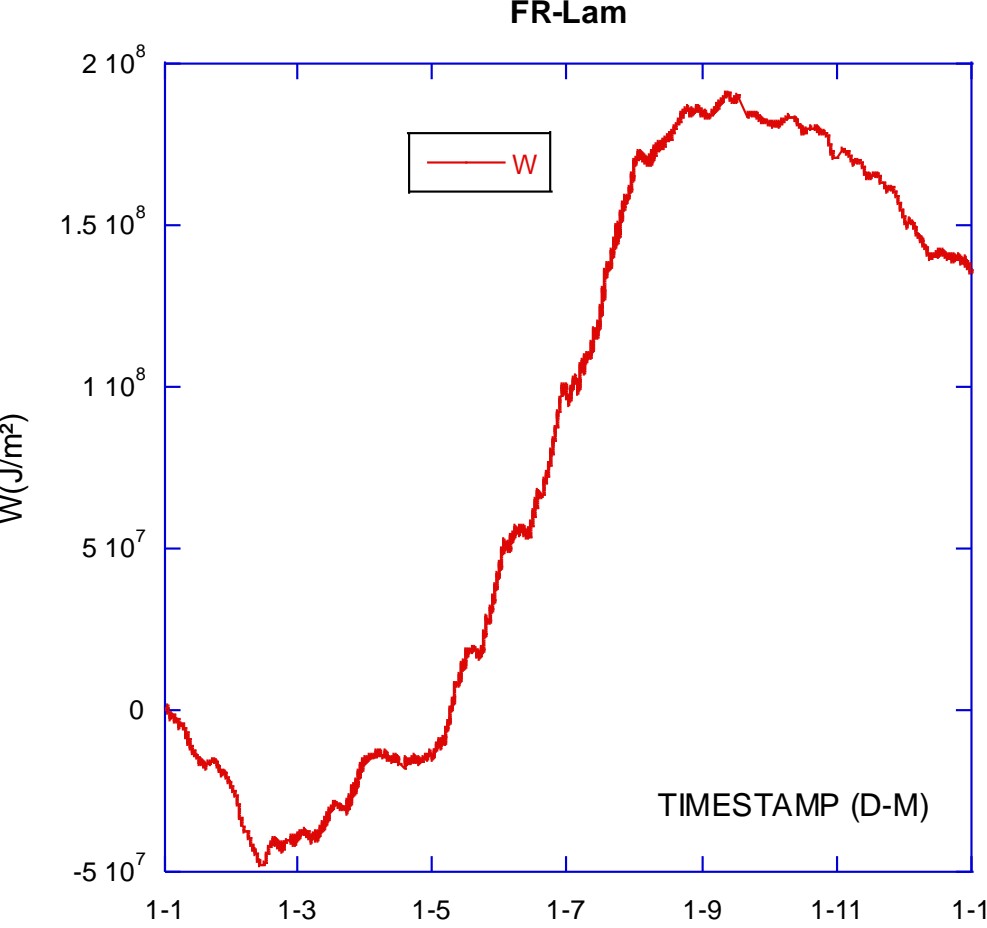

**Figure 2. Soil heat flux Integrated for one year.**

### 3.2.1 Sunshine or soil inhomogeneities.

An important imbalance may be induced by the soil surface inequal sunshine resulting in a non-uniform, direction-dependent, heat flux density. Making abstraction of heat storage above the flux plates and a possible non-uniform soil heat capacity below the plates, we can consider a simple limited shadowed surface case.

Figure 3 depicts a partially shadowed soil surface with three SHFPs. Plate A is installed on a sunny surface far from any shadowed surface. Plate B is installed under a sunny surface but close to a shadowed surface and plate C is installed under a shadowed surface. During the daytime (Fig 3. a) plate A and plate B will sense the same amount of heat resulting from solar heating. Plate C is installed under a shadowed surface, only a little heating is sensed by this plate. Bellow plate A, the soil is constituting a heat storage $S_A$ with all the heat penetrating the soil. Below plate B, one part of the penetrating heat is going under the near, shadowed surface as the soil is over there colder and only a part of the total heat sensed by plate B is stored as $S_B$. Below plate C, only a weak heat is penetrating the surface and the storage $S_C$ is constituted from this heat raised by the heat coming from the near sunny surface. We have then a relation:

$$S_A > S_B > S_C$$

(3)

In, the case of a relatively small shadowed surface we can even assume $S_B = S_C$. At night (Fig. 2.b), the soil below plate A is giving back the heat drawing from the storage $S_A$. The same for the soil below plate $S_B$ and $S_C$. However, the heat flowing up will be proportional to the corresponding heat storage and equation 3 is also valid for nocturnal heat effluxes. Then, the daily balance of plate A will be close to zero, B plate balance will be positive and C plate balance negative. Of course, if plate B is placed at a "symmetrical" emplacement of plate C, the positive daily imbalance of plate B is then opposite of C plate imbalance, averaging these two plates will recover the accurate measurements. This is one of the reasons to have numerous plates installed. However, a common behavior would push us to not install plates under a shadowed surface. Furthermore, this imbalance case is also valid for the coldest soil location due to a higher soil water content (Cabidoche and Voltz, 2005), especially in clayey soil. Indeed, if the soil surface is not perfectly flat or cracked, after a consequent rainfall and possible runoff (Novák et al., 2000) the rainfall water will naturally concentrate in all surface hollows and cracks.

These hollows or cracks will become colder than the rest of the soil and a natural underground heat transfer will attempt to equalize soil temperatures creating corresponding SHFP measurement imbalances. Non-uniform evaporation (different textures or cracks) creates also non-uniform soil temperatures. A non-uniform soil heat capacity (non-uniform density) is causing also in-depth heat exchanges. During the day, soil heat fluxes tend to

rise (vertical fluxes) and equalize soil temperatures (non-vertical fluxes) while during the night, the soil cooling is mainly resulting from a radiative exchange following Stefan-Boltzmann law:

$$M = \sigma \in T^4$$

(4)

With $M$ being radiant emittance (emitted energy per unit time per unit area), $\sigma$ is a constant, $\in$ is the soil emissivity and $T$ is the soil temperature.

Contrarily to the heat exchanges due to temperature differences this law is highly non-linear, then nighttime exchanges will not recreate daytime soil temperature inhomogeneities and resulting non-vertical soil heat fluxes

do not compensate for the daytime non-vertical soil heat fluxes. For this reason, for better representativity, SHFP shouldn't be placed in a vicinity of a pit dug for soil water content probes or any other artificial recent pit with an altered soil density nor in a vicinity of an abnormally compacted soil (enclosures) unless another plate is placed on the other side of the inhomogeneity to compensate the imbalances. In general, any soil temperature difference will give rise to below surface non-vertical heat exchanges creating surface heat flux imbalances. These imbalances

are positive and negative depending on which side of the inhomogeneity boundary is located in the measuring SHFP. By energy conservation, the real overall imbalance is nil. This point is very important as for the correct special representativity the plates should be placed on both sides of the inhomogeneity boundary measuring on both sides for a correct inhomogeneity representation. The overall measurement, averaging measurements of all the plates around an inhomogeneity, should display a nil imbalance.

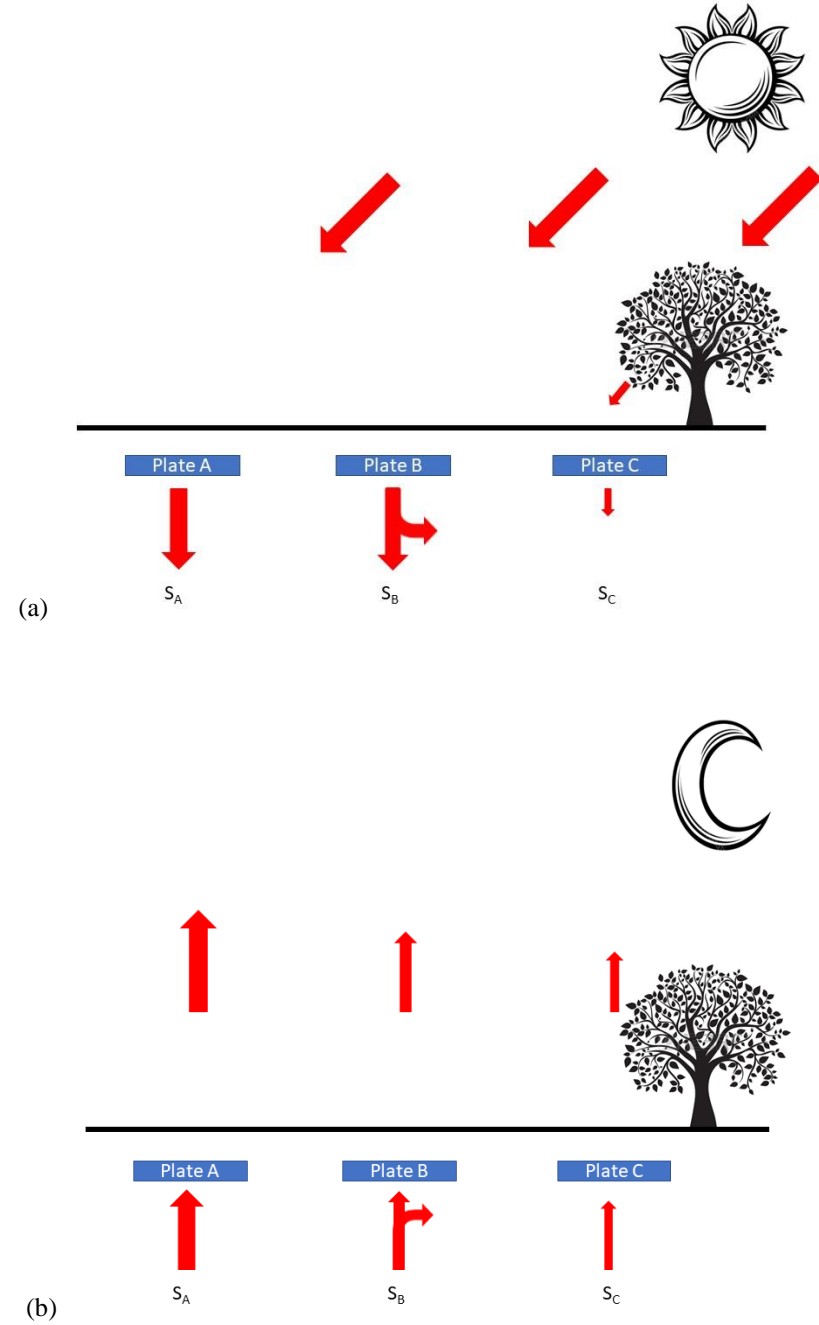

**Figure 3. (a) Daylight resulting heat flux on a sunny surface A with resulting Heat storage $S_A$, sunny surface B (Storage $S_B$) with close shadowed surface C (Storage $S_C$). (b) Nighttime heat flux resulting from heat storage emptying.**

For example, considering the previously depicted partially shadowed surface, and supposing that we have only two plates installed on this surface. If it is plate A and plate B, then the overall heat flux imbalance will be positive. If it is plate A and plate C, the overall heat flux imbalance will be negative and, if it is plate B and plate C; the overall heat flux imbalance will be nil. Using annual integration, we can see immediately that plate A does not have any inhomogeneity boundary in the vicinity and that plate B and plate C are "symmetric". In the case where only two plates are used, by individual integration we can see if the inhomogeneity boundary is present and was correctly compensated by placing as many plates on one side as on the other side. Of course, the reality is a bit more complicated since not only one inhomogeneity may be present, and convective fluxes causing also imbalances. However, the convective fluxes discussed later in this paper are less localized and an overall imbalance is easily identified in the FR_Lam field-deployed plates.

We can expect to overcome imbalances due to surface soil inhomogeneities using numerous flux plates "judiciously" placed. A much better understanding of the observed soil heat integration imbalances would be given by a correct three-dimensional heat flux measurement and not only a one-dimensional measurement. Three-dimensional heat flux sensors were proposed by Domínguez-Pumar et al. (2020) for regolith (fine soil, or dust of planets without an atmosphere). To my knowledge, a three-dimensional soil heat flux sensor for terrestrial use does not exist yet. A quick but not cheap solution would be to borrow three plates: one horizontally and two others vertically orthogonally to each other. Any sensed horizontal heat flux reveals a close inhomogeneity boundary.

If we are assuming that the observed unbalance is mainly due to convective fluxes, a minimization of the corresponding systemic error may be attempted by the yearly based soil heat balance closure with a deduced statistical correction.

Considering only a field-deployed SHFP first we can integrate their measurements with an adequate $G_{TH}^L$ correction over a year. Based on the computed imbalance and its deviation from an overall imbalance, decide which plate is correctly representative and which plate is not (Fig. 4). Discard data from obviously non-representative plates (G42 and G51 in this example) and form the overall measurement with the remaining data. We have to note that the considered data soil's December temperatures were slightly cooler than the soil's January temperatures. Differences range from 2.5 degrees to 1 degree depending on the depth (2.5 degrees cooler at the surface and 1-degree cooler at 100 cm depth). Then the calculated heat flux imbalance does not correspond to the soil temperature variation and would be even bigger if the soil temperatures were the same at the beginning and at

the end of that year. The fact that there is a large, quasi constant, soil heat imbalance in all remaining measured locations, is suggesting that this imbalance is not resulting from inhomogeneities. We can then attempt to correct it by *convective* heat flux considerations.

Below are listed some of the convective fluxes that can also cause notable imbalances.

### 3.2.2 Soil gas exchanges.

The soil is exchanging gazes, mainly respiration: $CO_2$ coming from the soil and absorbed $O_2$, and subsurface evaporation/condensation. For respiration, due to the characteristic heat capacity difference of $CO_2$ and $O_2$, we may also expect an energy exchange. This is the case but the total amount remains negligible (yearly about 100

J/m² for winter wheat culture).

The heat conversion from sensible to latent heat arising below the plate bias balance as the corresponding upcoming (or downcoming in the case of condensation) energy (latent heat) is not sensed by the plate, however, is still sensed by the air phase $L_e$ sensors such as eddy covariance setup.

The subsurface evaporated or condensed water is then added to the surface evaporated or condensed water when the corresponding energy was already (in the case of subsurface evaporation) or will be (in the case of condensation) accounted for by the soil heat flux plate measurement as sensible heat before or after the conversion. It is a *double-counting* as highlighted by Ochsner et al. (2006). Nota bene, the reality is even more complicated as the water vapor created or condensed under the plate may need some time to emerge or infiltrate from or into the

soil. The sensed water vapor in the air is then not only with multiple origins or pits but also with multiple conversion times complicating SEB closure.

In our case, the positive imbalance may be, in part, due to the below plate evaporation. As the plate is buried in a high clay content soil, the desiccation cracking may allow deep soil evaporation (Selim and Kirkham, 1970).

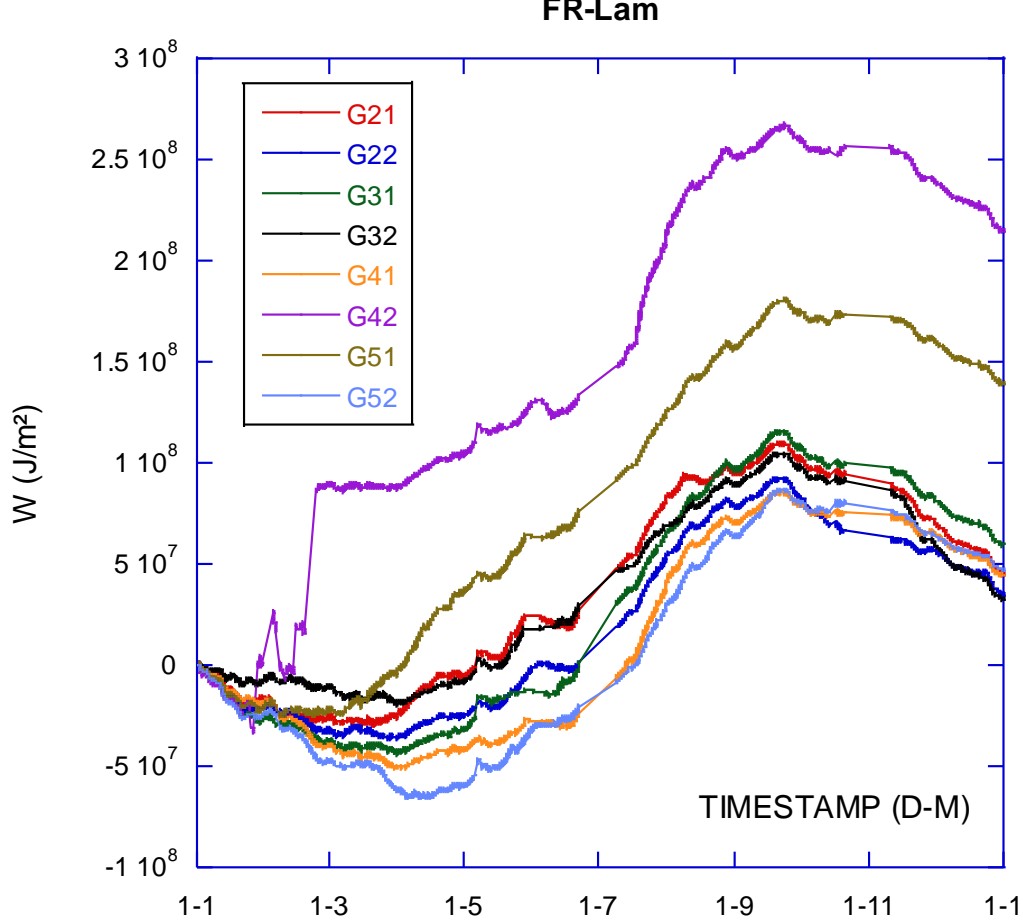

**Figure 4. Integrated raw measurements of eight heat flux plates installed on FR-Lam on an agricultural plot (cropland).**

### 3.2.3 Evapotranspiration.

A question remains open: except for a latent heat conversion below the SHFP is there another possibility to cause the soil heat flux imbalances? For example, the water absorbed by the roots is routed to the leaves and evaporated chiefly during the daytime and the hot seasons. This water migration is similar to convection and is not sensed by any heat flux sensor. Moreover, during the hot seasons, the deep roots absorbed water has a lower temperature than the soil surface temperature.  To equalize its temperature with the surrounding soil a heat transfer takes place lowering the soil temperature then lowering the soil heat storage and accentuating the heat transfer from the soil

surface. Figure 5 depicts the water absorbed by the wheat roots, flowing through the vegetable body and evaporating by the leaves.

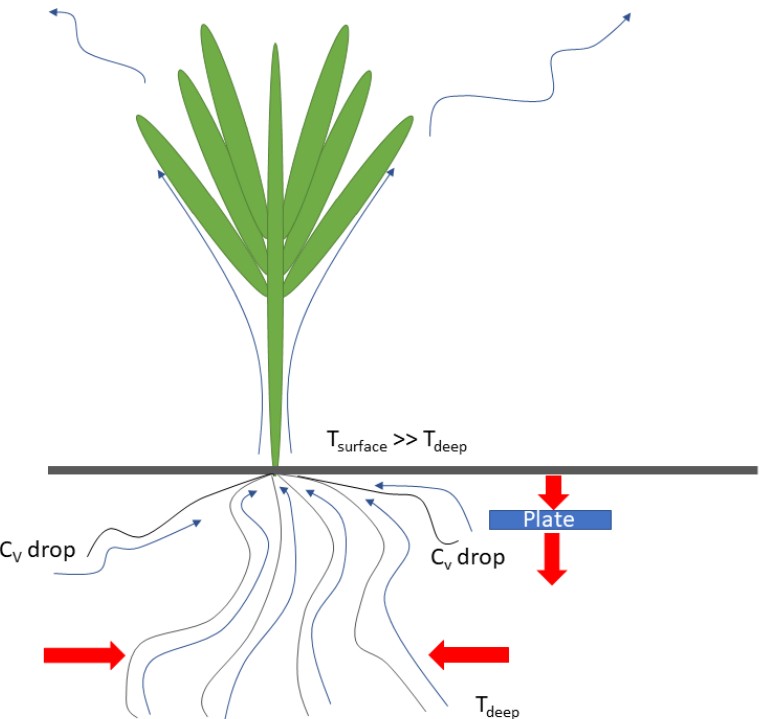


**Figure 5. Root absorbed water is flowing up from the deep soil at low temperature to the hot sun heated soil surface provoking a heat transfer between the soil and the roots. Shallow roots absorb water drying soil and lowering its heat capacity $C_V$. Water is storing then energy is evacuated from the soil.**

Even if the root absorbed water coming from the shallow soil layer, as water is an important part of the soil heat storage due to its high heat capacity, the daytime dried soil's heat capacity drops and, by nighttime, the soil is not able to counterbalance the daytime heat flux as the storage is not only a question of temperature but also a question of heat capacity. The water absorbed by the root, with corresponding stored energy, is not sensed by SHFP and there will be a resulting positive unbalance as the water stored energy is no more available for nighttime opposite

transfer. In general, any mass flow from beneath the SHFP, gaseous, liquid, or solid, will give rise to an energy evacuation and then heat flux imbalance. Considering the winter wheat daily water usage, the soil water table (assumed as only one source of the root absorbed water as winter wheat roots may reach over two meters depth

(Thorup-Kristensen et al., 2009) and the temperature difference with SHFP level soil temperature, a very rough estimation of the energy withdrawn from the soil bellow SHFP, gives an imbalance of about 20 MJ/m² a year for winter wheat (the culture of the considered year on FR-Lam). The assessed imbalance source is then comparable to the geothermal correction (see Sect. 3.2.5) with an opposite sign, and cannot explain alone the observed imbalance in Fig. 4 (50 MJ/m²). However, this estimation is certainly underestimated as the transpiration takes place mainly during the daytime when the temperature gradient between the soil surface and the deep soil is much more important than during the night. Then, the daytime deep soil water evacuation withdraws more energy than during the night and the daily average of the transpiration is underestimating that energy. Also, during the bare soil period, surface evaporation is forcing the soil water to migrate from the deep layers to the dried shallow layers. This migration is not sensed either by SHFP and adds a positive imbalance again rather for a long-term imbalance. The corresponding correction is noted $G_{ET}^{L}$. A similar mechanism causing soil heat flux imbalance is the soil water redistribution so-called *water lift* when some deep-rooted plants are pumping water from the deep wet soil layer and releasing it into the shallow dry soil layer due to the water potential $\Psi$ gradient (Horton and Hart, 1998). During the hydraulic lift, no evaporation is involved. Depending on how deeply is released deep-root pumped water, namely below or above the SHFP's level, the resulting convective flux may bias SHFP's measurement too. Note that only the beneath SHFP evaporation/condensation causes a *double-counting* problem.




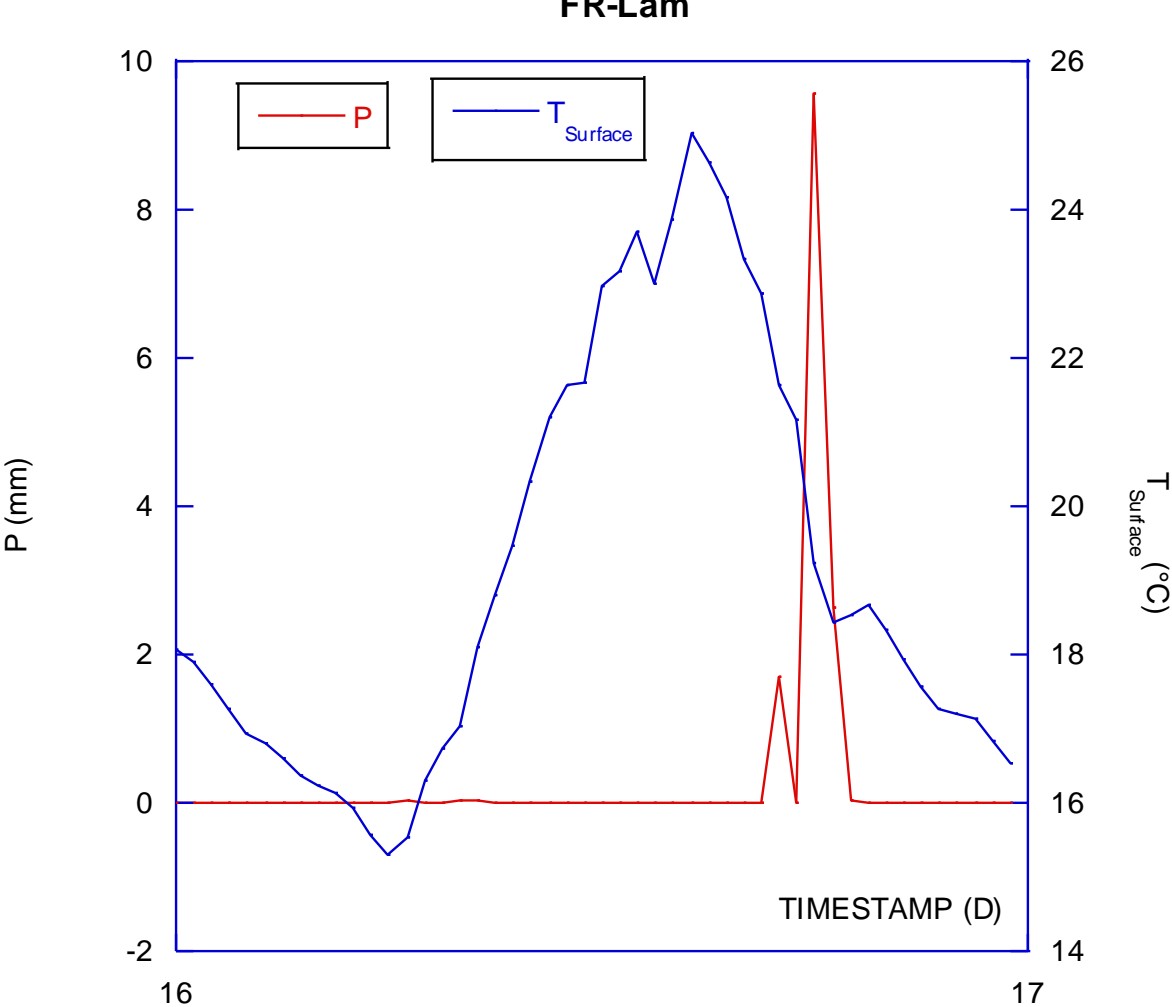

**Figure 6. Rainfall soil surface temperature cooling.**

### 3.2.4 Rainfall or irrigation is a negative and positive imbalance source.

On FR-Lam, the main water inputs are rainfall and irrigation. Other water inputs such as snowfall or hailfall are extremely rare.  Note that with the snowfall and hailfall energy apports would be more difficult to assess since
there is also heat absorption during later liquefaction.

The rainfall or irrigation P (in mm of water) is causing the soil surface cooling and provokes a negative soil heat flux (Fig. 6). This does not affect the SHFP balance (not at this stage, see further text) but the corresponding energy $H_p$ needs to be included in the SEB equation (see equation 8) as it is an external frigories apport proportional to rainfall intensity $P_I = \frac{\delta P}{\delta t}$, to the water heat capacity $C_w$ and the difference between falling water temperature $T_w$ with the soil surface temperature $T_s$:

$$H_p = P_I * C_w * (T_w - T_s)$$

(5)

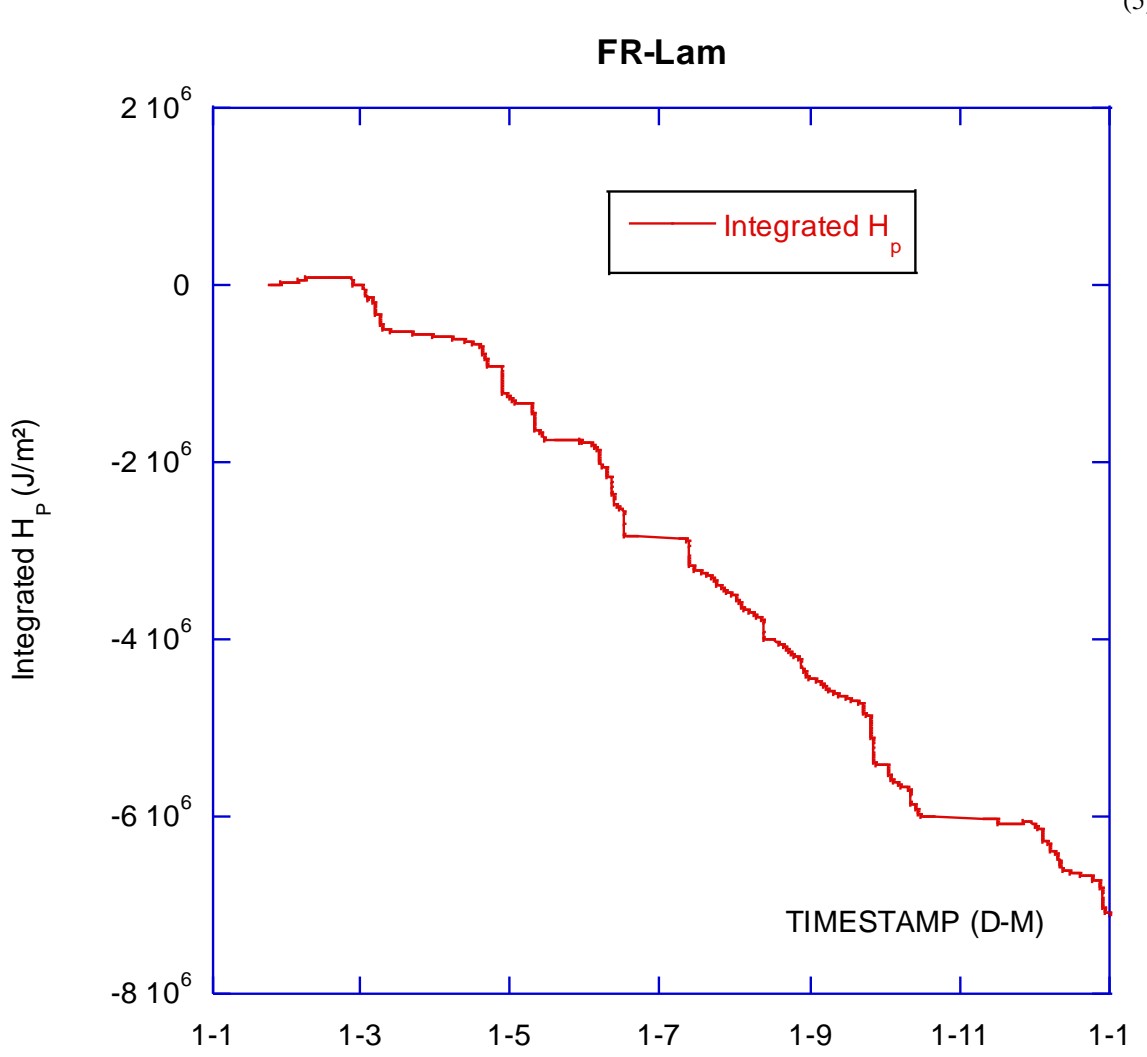

Figure 7. Integrated rainfall cooling $H_P$.


Unfortunately, we do not have any instrument installed on FR-Lam that can provide us with a rainwater temperature. As a rough approximation, the air temperature is used assuming that the falling water has the same temperature as the ambient air (this assumption is not valid for irrigations and overestimates water temperature for natural precipitations). After one year of precipitation, we obtain -7 MJ/m² (Fig. 7) which is not negligible on the

annual scale. On the short scale, the rainfall soil cooling is very important and the corresponding SEB is greatly affected (considering data shown in Fig. 6, cumulated rain cooling energy is $E_P =$ -289 kJ/m² and SHFP measurements show that when it would be about -10 W/m² of heat flux without the rain, it was -70 W/m² with the rain).

The rainfall (or hailfall) also brings energy through its high kinetic energy important enough to be considered an

important soil erosion factor (Wischmeier and Smith, 1994). Unfortunately, we do not have yet any disdrometer installed on FR-Lam making it difficult to assess the kinetic energy importance.

When the rainfall water is on the soil surface the SHFP measurements are not yet not imbalanced. Afterward the rainfall water is penetrating the soil and, similarly to evapotranspiration, SHFP is not sensing this migration but an important heat transfer by convection may take place (Kollet et al., 2009). This time the imbalance would be

negative if the infiltrating water was hotter than the deep soil bringing some calories. This happens when the soil surface temperature is higher than the SHFP level soil temperature (5cm on FR-Lam). This is not always the case especially at nighttime and event by the daytime during cold seasons.

The resulting heat flux $G_P^S$ would be similar to $H_p$ but using the difference in the soil surface temperature $T_S$ and

the SHFP level soil temperature $T_5$.

$$G_P^S = P_I * C_w * (T_S - T_5)$$

(6)


 Figure 8 depicts the cumulated $G_P^S$ . We can note that after one year the results are almost nil, under 0.022 MJ/m². Then, we cannot assume the rainfall water convection counterbalances the evapotranspiration water convection for SHFP measurements on a long-term scale on FR-Lam. With nighttime irrigation, results would be positive, and with daytime irrigation, results would be negative but if the irrigation is limited then the overall additive would

be limited too however, à short-term correction may be necessary.

All these considerations may deserve more investigation work.

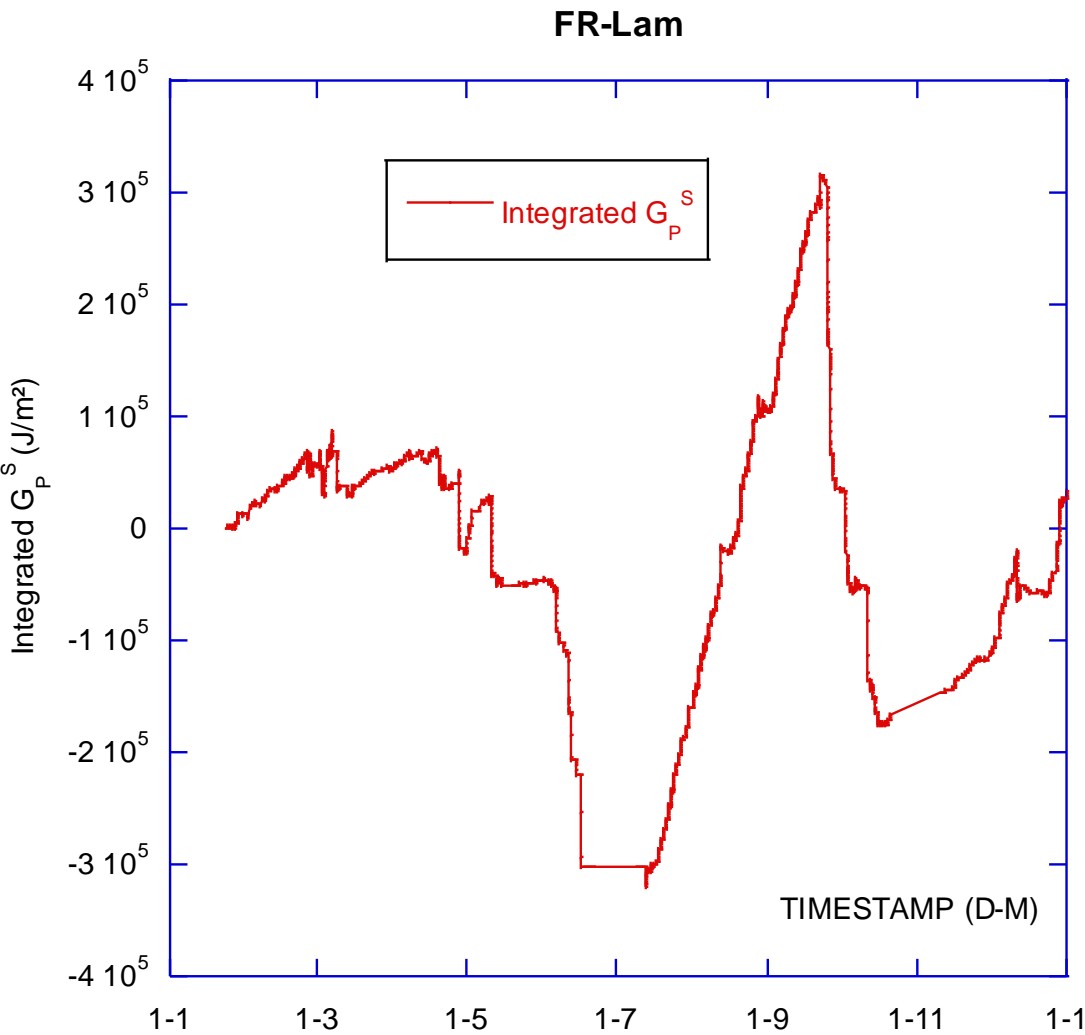

**Figure 8. Integrated factor of precipitation with soil surface temperature difference with soil 5cm depth**
**temperature.**

**3.2.5 Geothermal heat flux.**

Concerning the geothermal heat flux, well sensed by the SHFPs, even if $G_{TH}^L$ is relatively small in respect of the solar maximum radiation and the nocturnal soil maximal heat efflux, this heat flux is always upgoing. At the same time, when totalizing energy fluxes, as solar radiation heating is counterbalanced by nocturnal soil radiation, the diurnal and especially the annual imbalance due to the geothermal heating flux may be important (Fig. 9). Consequently, a geothermal correction is rather for a long-term integration check.

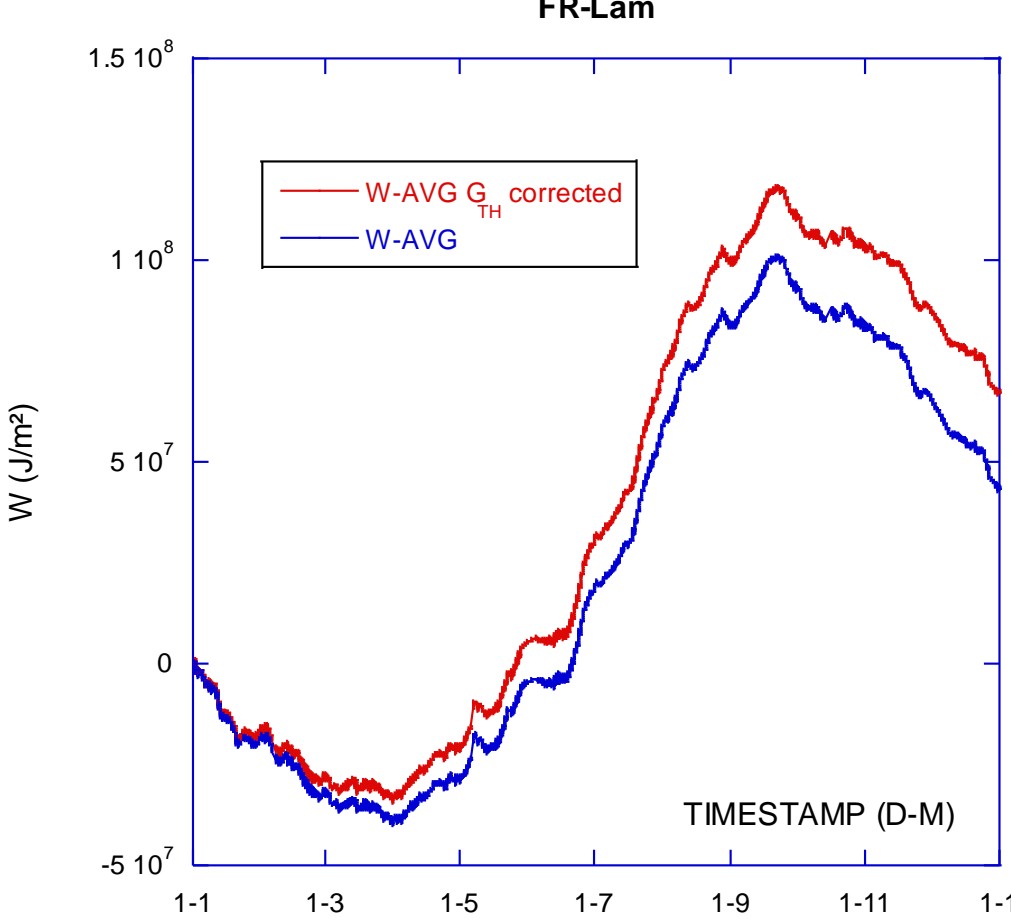

**Figure 9. Integrated averaged, among the plates, measured soil heat flux: W-AVG and the same integrated flux with geothermal efflux subtracted: W-AVG $G_{TH}$ Corrected.**

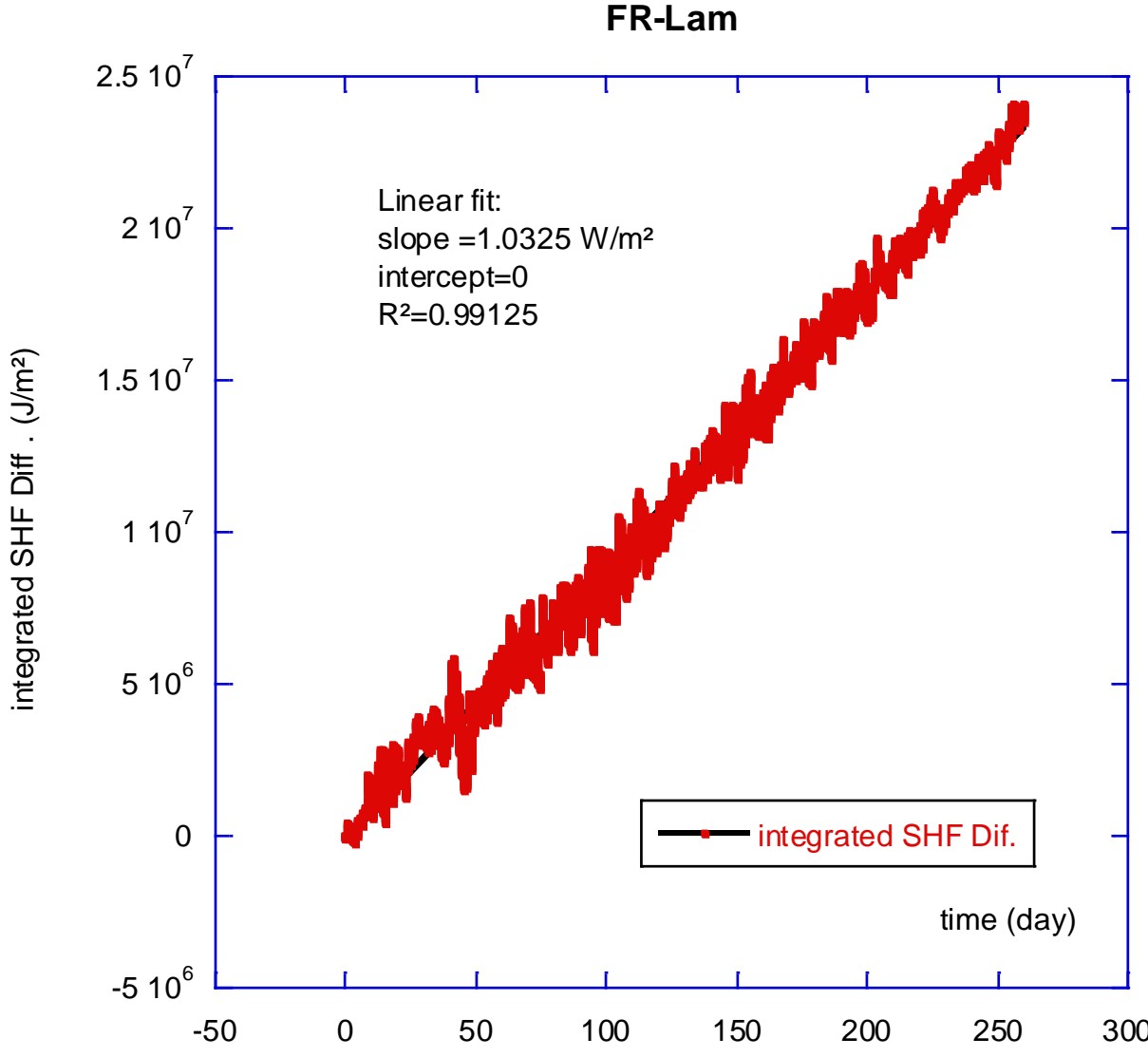

**Figure 10. Integrated SHF Diff. along with linear regression.**


### 3.2.6 Calibration data.

There is also a well-known, but deserving to be signaled again, the precaution that should be taken when working with the self-calibrated flux plates. Because during the calibrations an artificial heat flux is generated, during and one hour, or even more, after the calibration the initialization data have to be discarded. Not only the generated heat is sensed but also the surrounding soil is heated and needs time to cool down. If corresponding data are not discarded an overestimation of the heat flux is observed. It is less known that for the committed error, when not discarding calibration period data, a rough correction remains possible. Figure 10 shows the integrated difference (SHF Diff.) between measurements with all data including calibration periods and measurements where, during and one hour after calibration, the data are discarded.

*SHF Diff = (Half-hourly averaged measurements with all data available) – (half-hourly averaged measurements with discarded data during and one hour after calibration).*

(7)

As we can see, this difference integrated over time is following a straight line which means the average heat flux measurements, with calibration data, can be corrected with a simple additive: -1.0325 W/m² in our case, with rather good accuracy ($R^2 > 0.99$). It is consistent with the calibration process as the total applied heating is 1.4 W for 4 minutes every 7 Hours then averaging this heating power along with SHFP diameter (80 mm) gives an average of 2.65 W/m².

### Conclusion.

Self-calibrated SHFPs are probably the most used sensors for *G* measurements. This technique is reliable however, important errors that are not always taken into account may bias the results. Some of the errors are avoidable, others result from physical phenomena and may still be present even if all the precautions are undertaken. It is important to carefully check the installation place considering a possible imbalance by an annual integration. The annual integration allows to check quickly each SHFP, individually, and to select representative plates based on an obvious divergence of an observed annual imbalance versus overall annual imbalance. This way is very easy to compute and allows an immediate sight check contrarily to the non-integrated soil heat fluxes results. In case of a systematic relative imbalance of all plate measurements, a statistical correction may be attempted. A beneath SHFP water evaporation and other phenomena such as evapotranspiration or rainfall, or any water infiltration, may contribute to the sensed heat imbalance.

Concerning the SEB equation (Eq. 1), since SHFP are sensing only the conduction heat flux, the $G$ term should also include corrections for short- or long-term measurements such as $G_{ET}^L$ or $G_P^S$ and other terms such as rainfall or irrigation, snowfall, hailfall, but also mist and fog (Yin and Arp 1994), dew (Jacobs et al., 2006) or marine breeze (Drobinski et al. 2018) $H_P$ which should be added as these energy fluxes are not negligible when totalizing energy variations and do not originate from solar or resulting heat flux may be sensed by flux plates or other heat flux sensors. Assuming appropriate inhomogeneities influence compensation and the beneath plate evaporation negligibility, the SEB equation becomes:

$$R_n - (G^C - |G_{TH}^L| - G_{ET}^L - G_P^S) - (S_C + S_P) + H_P = H + L_e$$

(8)

Here, as mentioned previously, by simplification, $G^C$ contains the below SHFP heat storage. Please note that we have to add (or subtract the absolute value as $G_{TH}^L$ is negative) the $G_{TH}^L$ into the SEB equation only if a geothermally corrected, for convective fluxes assessing purpose, sensed soil heat flux $G^C$ is used as the SHFPs are well sensing the geothermal conductive heat flux and as this heat flux is real. In the case of a geothermally non-corrected SHFP's sensed heat flux using $G$ in the SEB equations, we do not have to add the $G_{TH}^L$. Note also that all the corrections on $G$ or $G^C$ are not helping to solve SEB closure problems when using the eddy covariance technique for $H + L_e$ measurement as these corrections tend to lower sensed $G$ or $H + L_e$ are usually already too small for SEB closure (over 30% disclosure on FR-Lam (Dare-Idowu et Al., 2021)) suggesting that the eddy covariance technique sensibly underestimates $H$ and $L_e$ measurements. Only the $Hp$ term helps for SEB equation closure as it represents mainly a soil surface cooling, then a negative term. The vegetation heat storage and photosynthetic activity may be added to complete this equation.

For better energy transfer monitoring, I'll suggest measuring not only the water table depth but also the soil water table temperature and the rainfall water temperature for further calculation.

**Appendix A.**

SHFP measures a punctual vertical conductive heat flux. Punctual, because the measuring surface of the SHFP is very small compared to the eddy covariance footprint. This appendix describes the one-dimensional heat flux and the annual integration nullity explanation.

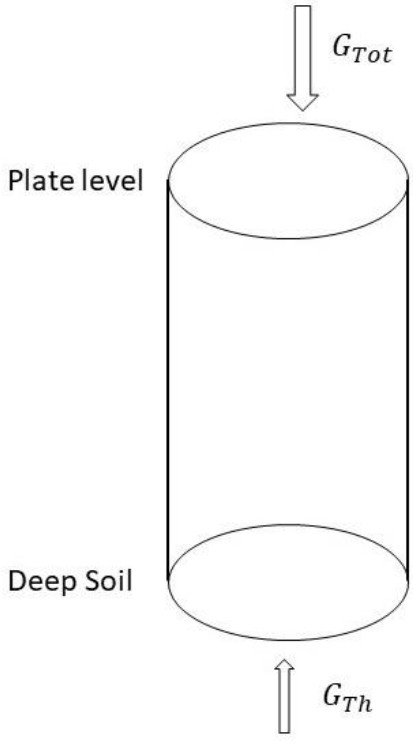


**Figure A1. Soil colon between the soil surface and a deep soil level where the soil temperature does not change during the year.**

Let's consider a homogeneous soil column between the SHFP depth and the depth where the soil temperature is
invariable during the year (Fig. A1). Internal energy $E$ conservation of an inert core law, inert means exempt of endo or exo-thermal chemical or physical reaction, allows us to consider the integration of all heat exchanges of this core with the surrounding environment. Indeed, the fundamental energy conservation law can be expressed as

energy variation $\Delta E$ equal to the temporal integration of the heat densities exchanges integration around the core surface:


$$\Delta E = \int_{t_0}^{t_1} (\oiint dG)dt$$

(A1)

In the case of a homogeneous soil with a virtually delimited colon, the lateral heat exchanges are nil and only the heat exchanges located at the lower surface or the upper surface are not nil. In other words, through this soil column, one-dimensional heat flux is entering or quitting by the upper side and by the lower side.


$$\oiint dG = S(G_{Tot} + |G_{Th}|)$$

(A2)

S being the top and bottom surface of the colon.

In practice, the SHFP depth is 5cm and we can consider the soil temperature as invariable at 1000cm depth. At the top, the incoming heat flux is $G_{Tot}$, and at the bottom, since the surface heat flux variations were absorbed through

the soil column, there is only the upgoing geothermal heat flux coming from the deep soil and resulting $G_{Th}$. For clarity, as the geothermal heat flux is upgoing and then, following the conventions adopted for the heat fluxes exchanges measurements, is negative, the absolute value of this heat exchange is considered.

The soil column stores thermal energy and its variation $\Delta E$ between two instants $t_0$ and $t_1$ can be calculated by integrating entering or quitting heat flux from the top and the bottom:


$$\Delta E = S \int_{t_0}^{t_1} (G_{Tot} + |G_{Th}|)dt$$

(A3)

If we assume that after one year the soil temperature profile and specific soil capacity profile did not change, it means there is no energy variation stored inside the considered soil column, then the energy variation $\Delta E$ is nil.

From equation A3 we obtain then sensed annual heat soil surface flux nullity after geothermal heat flux subtraction (or its absolute value addition):

$$\int_0^{365} (G_{Tot} + |G_{Th}|)dt = 0$$

(A4)

Using an SHFP as a sensor for $G_{Tot}$ measurements, the non-nil results of the annual integration represent the imperfection of the SHFP measurements.

These imperfections could have two distinct origins: inhomogeneity boundaries causing non-vertical, lateral, heat exchanges (one-dimensional heat flux does not apply anymore) and not sensed convective heat fluxes. For

convective fluxes exchanges important example please see appendix B.

The geothermal heat flux subtraction (or its absolute value addition) is proposed for the missing heat flux parts estimation considering the resulting annual integration nullity.

**Appendix B.**

The heat fluxes exchanges are composed of three different components:

-   Conductive fluxes dues to a contact of two corps with different temperatures where the hotter corps give in thermal energy to the colder corps. These exchanges can be measured by temperature measurements across a well-known third corps such as SHFP.

-   Radiative fluxes, where the corps are losing or receiving energy by radiation according to the Stefan-

Boltzmann law. It is important to note that radiative exchanges concern only the surfaces on a sightline provided they are at different temperatures. These exchanges can be measured by radiation sensors, mainly on the infrared scale when the usual temperatures are concerned.

-   Conductive fluxes due to the fluid's movements where fluids, gases, or liquids, move displacing with them thermal energy proportional to the mowing fluid quantity, its temperature, and specific thermal

capacity. These thermal exchanges are very difficult to measure since the moving fluid quantity should be measured along with its temperature and specific thermal capacity.

In some cases, the convective fluxes may be preponderant. For example, in a room of standard height of 2.6m with operating at high-temperature underfloor heating and covered by a poorly insulated roof, therefore with

a cold ceiling, it is well known that the air heated by the floor (lower density) migrates upwards to accumulate under the ceiling pushing the cold air (higher density) downwards. It is typical *convective* heat exchange.

Thus, the air with the highest temperature accumulates under the ceiling. If we place an SHFPS at a height of 2m, it will indicate a heat flow from top to bottom because the temperature at the top is higher than that at the bottom, and the SHP's measurements correspond to the *conductive* heat exchange at 2m height. If we rely solely on the indications of this SHFP, we come to an absurd conclusion that the heat exchanges go from the cold ceiling to the warm floor. This comes from the fact that it is necessary to consider all the heat exchanges, which include the *convective* exchanges dominating the conductive exchanges in this example due to the high air density temperature dependence and high air mobility. The SHFP are measuring only the conductive part of the heat exchanges. The radiative part of the soil heat fluxes exchanges take place on the *soil surface* and is measured by the net radiometer but the convective fluxes into the soil are usually neglected.

**Code and data availability.**

The data and source code used for these studies can be obtained by contacting the author.

**Competing interests.**

The author declares that he has no conflict of interest.

**Acknowledgments.**

I would like to thank the technical team and more particularly Franck Granouillac CESBIO who is greatly contributing to the installation maintenance.

**Financial support.**

This project was funded by the Institut National des Sciences de l'Univers (INSU) through the ICOS ERIC and the OSR SW observatory. Facilities and staff are funded and supported by the Observatory Midi-Pyrenean, the University Paul Sabatier of Toulouse 3, CNRS (Centre National de la Recherche Scientifique), CNES (Centre National d'Etude Spatial), INRAE (Institut National de Recherche pour l'Agronomique et Environnement) and IRD (Institut de Recherche pour le Développement).

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
