# Peer review of "The Soil heat flux sensor functioning checks, imbalances'"

_Geoscientific Instrumentation, Methods and Data Systems, 2021_

## Author Comment (AC2)

Let's consider a homogeneous soil column between the SHFP depth and the depth where the soil temperature is invariable during the year. Through this column, one-dimensional heat flux is entering or quitting by the upper side and by the lower side.

In piratic, the SHFP depth is 5cm and we can consider the soil temperature as invariable at 1000 cm depth. At the top, in the ideal case, the SHFP measures the total heat flux: $G_{Tot}$, at the bottom, since the surface heat flux variations were absorbed through the soil column, there is only the geothermal heat flux coming from the deep soil: $G_{Th}$.

This soil column stores some thermal energy and its variation $\Delta E$ between $t_0$ and $t_1$ can be calculated integrating entering or quitting heat flux from the top and the bottom:

$$\Delta E = \int_{t_0}^{t_1} (G_{Tot} - G_{Th})dt$$

If we consider that after one year the soil temperature profile and specific soil capacity profile did not change, it means there is no energy variation stored inside the considered soil column, then the energy balance should be nil:

$$\int_{0}^{365} (G_{Tot} - G_{Th})dt = 0$$

The non-nil results of this integration represent the imperfection of the SHFP measurements.

These imperfections could have two distinct origins: inhomogeneities boundaries causing non-vertical, lateral, heat exchanges (one-dimensional heat flux does not apply anymore) and not sensed convective heat fluxes.

The geothermal heat flow subtraction is proposed for the missing heat flow parts estimation.

---

## Author Response (AR1)

Dear Editor,

I changed the manuscript to take into account the two referee comments and suggestions. The new organization is slightly modified to make this paper clearer. I am adding an appendix with an explanation of the annual integration that should be nil. I am adding also mentions and references suggested by the second referee about other possible water imports and movements.

The use of the word "flux" versus the word "flow" is a little bit confusing in the existing literature but I tend to uniformize this use in the text reserving "flow" for mass or energy flow and flux for its measurement. Consequently, some sentences, including the title, changed.

Best regards.

Bartosz Zawilski

Answer to the referees.

I am sincerely grateful to Leonardo Montagnani, designated hereafter as "RC1", and particularly to the second referee designated as "RC2", for the comments, criticism, and enriching suggestions that helped me to improve this paper.

I would like to answer, point by point, the first referee RC1 including the answer that I have alredy posted during the interactive discussion.

RC1: « The paper is relevant since it poses a basic question related to G measurements. In the end, what is the uncertainty related to G measurements? Should we add other heat flux plates, since there is a problem of spatial heterogeneity, or the overall accuracy and precision are too low, so G measurements are nearly useless, and it would be better to concentrate the efforts on multiple net radiation instruments instead? An answer to this question would be relevant in the flux community. »

The uncertainty of the G measurement is very site-dependent. Depending on the soil properties, water table depth, vegetation cover, pluviosity, and so on. The main uncertainty is not coming from measurements errors but rather from technique limitations and, more particularly, from *conductive* heat exchanges versus *convective* heat exchanges. To my knowledge, there is no sensor able to measure soil *convective* heat fluxes. Then, we cannot just measure, we have to assess the unmeasured heat fluxes and correct them, if possible.

I am stating L181-:" On can expect to overcome imbalances due to surface soil inhomogeneities using numerous flux plates judiciously placed." Adding more sensors to overcome spatial variability is always desirable, but not always feasible. The practical aspect needs to be considered too. All sensors need to be purchased, installed, interfaced, maintained, frequently checked, and so on. Unfortunately, it is not just about making a probability calculation.

The overall precision of the soil heat flux measurement, in my mind, is not a disqualifying problem, and G measurements are necessary. The "errors" are not exactly what I would call "measurement errors," but rather what I would call "technical errors." This means that the used technique inevitably represents only a very *local conductive* heat flux and does *not measure every overall* heat flux. Consequently, I would rather talk about the *leak* of measurements and *omissions*. Namely, when we are measuring soil heat flux using the heat flux plates, we are measuring local, one-dimensional, *conduction* heat flux neglecting local *convection* heat flux and horizontal deviations. This means that even if the conduction heat flux is spatially representative and exempt from any calibration error the convection heat flux is ignored. And the corresponding unbalance is not negligible on FR-Lam. Also, the target is to measure the overall heat flux as the eddy covariance footprint is much larger. SHFP measuring surface is very small making its measurement very vulnerable to local inhomogeneities influence. Assuming SHFP measurements are representative, we have to assume that the local inhomogeneities are correctly measured, which means on both sides of each inhomogeneity.

On the other hand, concerning the surface energy balance, not only solar radiation and geothermal energy have to be considered but all the energies brought to the soil surface such as rainfall. Snowfall and hailfall energy apports are more difficult to assess since there is also heat absorption during later liquefaction. This point was not raised in my paper since snowfall and hail-fall are extremely rare on FR-Lam but the corresponding sentences mentioning also mist, fog, marine breeze, and dew, as suggested by the second referee, are added to the revised version.

RC1: "One problem quite evident is the language. I am not a native English speaker, but I notice several basic errors (like a singular verbal form after a plural subject) that should be corrected."

Although I agree that my English writing is not my strongest point in this paper, I plan to have a native speaker correct it.

RC1: "A second weakness is about a possible synthesis, which is lacking in my view. I propose to organize the different topics into groups based on the fact they are producing random errors (so the error tends to zero in the long term) or systematic or selective-systematic errors. See Moncrieff et al. 1996. »

I reorganized the revised version. Moncief et al. describe errors in carbon dioxide flux measurements by the Eddy Covariance technique. Measurements errors arise from non-respect for eddy covariance requirements, spatial inhomogeneities, and eddy covariance technique uncertainty. Eddy Covariance technique is complex, requiring special aerodynamic conditions which is not the case with heat flux plates. In my paper, I am pointing out the fact that spatial inhomogeneity causes positive or negative deviations which could be checked by annual integration for each plate because we are usually using several plates when eddy covariance setup is rarely duplicated on a site. I am pointing out also *convective* soil heat fluxes resulting from evapotranspiration (vegetation respiration and soil evaporation) mainly giving rise to positive $G^C$ imbalance, and from rainfall water infiltration giving rise to negative or positive $G^C$ imbalance. On FR-Lam the sum of rainfall water infiltration contributions is close to zero but I'll hardly call it "random error". I am also pointing out soil surface energy apports that are neglected

such as rainfall. Resulting errors are not caused by instrumental imprecision but by used techniques limitations or by measurement leaks such as convective fluxes measurement. Unfortunately, we do not have yet specific sensors installed on FR-Lam to complete the necessary measurements and I can only roughly assess by minimizing each contribution which seems to be all the same not negligible. These complementary instruments are not commonly used or even do not exist yet. In other words, as I said, I would rather talk about technical errors, which means measurements leak, rather than measurement errors. Some of these measurement leaks are important for short-time integrations such as rainfall water infiltration but not for long-time integration (on FR-Lam). Others, on the contrary, are important for long-time integrations but not for short time such as evapotranspiration. Others for both, short-time and long-time integration such as rainfall energy apport. Anyway, in my mind, the only *random error* I can see is the SHFP emplacement near an inhomogeneity without another SHFP installed at a "symmetrical" emplacement counterbalancing its deviated measurements. These random errors are minimized by multiplying the SHFP and by avoiding obvious artificial inhomogeneities where a "symmetrical" SHFP emplacement is not even always possible. The SHFP number is then the minimizing error factor, not the time. Some of the measurements are not relevant for a longtime integration on FR-Lam, because of the specificity of the local climate but may be necessary for another location. This statement is added to the revised version.

This paper presents three main problematics:

1) Soil inhomogeneities and possible measured $G^c$ unbalances. By placing multiple SHFPs away from artificial inhomogeneities, inhomogeneities imbalances can be overcome. An a posteriori check for obvious discrepancies can be used to eliminate SHFPs for a representative overall conductive soil heat flux calculation.

2) Convective flux assessment. Subtraction of geothermal heat flux and yearly integration followed by convective flux assessment (such as evaporation, evapotranspiration, rainfall water penetration). Convective heat fluxes are real and not sensed by SHFP which is a measurement leak and should be assessed and added if possible. Beneath SFHP evaporation is sensed by eddy covariance giving rise to a "double counting" which is an error and should be corrected if possible.

3) Neglected energies and their importance assessment. To bring the SEB equation into balance, these energies must be included.

RC1: "Another point on which I disagree is the use of the annual sum of G as a benchmark if is close to zero. This leads for example, to the suggestion of the removal of geothermal energy. In my view, all the soil heat flux plates located in the footprint of an eddy covariance tower should be representative of the actual energy flow and not corrected. I would place the sensors on the shade, in the partial shade and in full sun since the scope of the measurement is to assess the average value of the selected variable and its standard deviation »

I do agree with the point that all the plates *should* be representative of the actual energy flow. Unfortunately, they are not because they cannot be. Eddy covariance footprint is much larger than the SHFP area then, in a case of not-perfectly homogeneous soil, only one SHFP cannot represent overall soil heat flux for the eddy covariance measurements but, in the best case, the overall measured soil heat flux when SHFPs are located in the eddy covariance footprint which is far to be always the case. I think that one of the crucial points was not clearly expressed in my original paper. Indeed, all inhomogeneities are causing non-vertical heat fluxes which give rise to the SHFP measurements imbalances *only on the boundary* of these inhomogeneities and *the real, overall perturbation is nil*. We can see it as energy conservation. In other words, any inhomogeneity may perturb the local SHFP measured heat flux balance bat not the overall heat flux balance in the area containing this inhomogeneity if correctly measured with numerous enough plates. Depending on which side of the boundary of the inhomogeneity is placed the SHFP, its measurement *balance* will be positively or negatively influenced. In my mind, "adequately measured inhomogeneity" is then an inhomogeneity with SHFP placed on each side of the boundary giving overall measurement exempt from the inhomogeneity boundary influence because the real, overall heat flux is not imbalanced. Theatrically speaking it is possible to "adequately measure" all inhomogeneities with numerous SHFP but in piratic it is not evident to know where are the boundaries. Using numerous randomly placed SHFPs, the *probability* of valid representability is rising but this point is not guaranteed in any way and has to be checked. With only one-dimensional measurements, it is difficult, if not impossible, to correct any SHFP measurements concerning the non-representability of the overall measurement. The only way that I can see is to discard one or more implicated SHFPs if their imbalances are too far from the overall measurement imbalance. It would be possible to investigate closer SHFP measurements with 3D fluxes measurements but it is not yet the case. With vertical and non-vertical heat fluxes measurements inhomogeneities influence could be better delimited. This statement was added to the text. With the new overall measurement calculated using only approved SHFPs, a missing soil heat flux, see next point, can be assessed. Please, note also that we are placing the SHFP horizontally, assuming a one-dimensional flow, which means that only the vertical heat flow is measured. This assumption is no more valid in the case of shallow non-vertical heat exchanges, it means in the case of shallow inhomogeneity boundary presence. In this case, the SHFP measurement is no more representative of the actual overall energy flow. What is representative is the overall SHFPs' measurement with SHPS placed on both sides of the inhomogeneity boundary and this has to be checked. For example, using my scheme of three plates with one under a tree shade (Fig. 3): supposing that the plates are measuring the real heat flux. If plate C (under shadow) is placed symmetrically to plate B, it means that its imbalance is opposite to the plate B imbalance, then the overall heat flux imbalance is nil, as it should be because the imbalance of the real, overall heat flux present on the considered surface is nil. Now, supposing that we have only two plates, not three plates, installed on the same surface. If it is plate A and plate B, then the overall heat flux imbalance will be positive. If it is plate A and plate C, the overall heat flux imbalance will be negative and, if it is plate B and plate C; the overall heat flux imbalance will be nil. Using annual integration, we can see immediately that plate A does not have any inhomogeneity boundary in the vicinity and that plate B and plate C are "symmetric". In the case where only two plates are used, by individual integration we can see if the inhomogeneity boundary is present and was correctly compensated by placing as many plates on one side as on the other side. Of course, the reality is a bit more complicated since not only one inhomogeneity may be present, and not only inhomogeneities causing imbalances.

Second point: SHFP can only measure the conductive heat exchanges, and the resulting heat flux measurement does not represent the total, *real* heat flux. The geothermal heat flux $G_{Th}$ is sensed by the plates (G), and corresponding subtractions are suggested for an

unbalance check by annual integration ($G^c = G-G_{Th}$). Integration of SHFP's measurements allows us to compare their behavior more precisely. When all SHFPs, as they do on FR-Lam and FR-Aur, exhibit roughly the same $G^c$ imbalance, we can assume that the special variability is not to blame. In this situation, the possible corrections do not concern natural inhomogeneities but rather missing measurements. For a better explanation, I am adding an appendix with a simple scheme of a soil column along with a short comment. Please see the posted pdf for the corresponding scheme and formulas.

RC1: Line (L) 11: "'latent heat conversion…' into liquid water?"

Mainly by evaporation (into water vapor then), but condensation is also causing integrated $G^c$ imbalance when the condensing water vapor is coming from the atmosphere, not from the deeper soil. This may happen with the crack presence. If the condensing water vapor is coming from the deeper soil, before becoming water vapor it was liquid water, before a condensation, we have evaporation. In this case, condensation counterbalances the evaporation effect. The solid (ice) melting into the liquid water gives rise to energy absorption too.

RC1: L26" Many process->many processes ( I report a few examples only of grammar errors)."

Grammar and spelling corrected.

RC1: L46 "the plates are not a technique"

Indeed, the plate *uses* a technique of sensing the temperature difference between two faces of a well-known material crossed by a heat flux. To my knowledge, all the plates use the same technique. The corresponding sentence is reformulated.

RC1: L65 "'de Beeck'-> Op de Beeck."

Done. I apologize to Maarten Op de Beeck.

RC1: L49" 'biased by inhomogeneities'. As mentioned above, I believe that all inhomogeneities should be adequately measured in proportion to their contribution to the overall flux."

This point is effectively similar to the previous one raised at the beginning of RC1 comments. Again, the inhomogeneities are not influencing the overall measurement *imbalance* but only influencing the local measurements imbalance close to the inhomogeneities boundaries installed SHFP. The target is not to correct the concerned measurements but to check that we have as many positively influenced SHFP as negatively influenced SHFP and to discard if any, obviously biased SHFP for overall heat flux calculation. This is suitable for later assessment of missing soil heat fluxes.

As mentioned previously, with a preserved soil temperature profile and preserved specific heat profile after one year, the overall flux integration, once geothermal energy is subtracted, should be nil, by definition. But it is true if, and only if, the plates' emplacements are representative. It means if all SHFPs are placed to counterbalance inhomogeneities deviations since each inhomogeneity gives rise to a positive imbalance in one emplacement and a negative imbalance in an adjoining emplacement the sum

being nil by energy conservation. If not, the annual integration of the overall measurement is not nil and the main problem is then the measurement *representativity*. We cannot talk anymore about adequately measured inhomogeneities. Furthermore, we never know in advance if plates are adequately measuring each inhomogeneity in proportion to their contribution to the overall flux (which is nil). Statistically, the more SHFP we are placing, the better is the chance to get representative measurement but it is not guaranteed. For these reasons also an annual integration of each SHFP measurement and the overall measurement gives us a quick idea about each SHFP representability and overall missing heat flux measurements.

RC1: "…11.2% sand, 2.8% organic matter. Here, granulometry and chemical composition are mixed up."

Absolutely, this soil classification is realized according to Malterre H. and M. Alabert (1963), Nouvelles observations au sujet d'un mode rationnel de classement des textures des sols et des roches meubles - pratique de l'interprétation des analyses physiques. Bulletin de l'AFES 2. pp. 76-84. Corresponding reference added into the text.

RC1: "L91-92: This sentence is explained in the following paragraph only."

Indeed, the corresponding indication is added to the concerned sentence for more clarity.

RC1: L97"'glocalization': geographical location?"

Mistyping, the correct word is "geolocalization" or Geo-localization.

RC1: "Around L 165 (Figure 3)., but why not use the data coming from partial shade? In a savanna, should all the trees be avoided? If we perform a stratified sampling, all the strata should be sampled. If we use random sampling, why not measure at specific, randomly selected, locations? I believe that this reasoning introduces a bias, not the contrary."

In this paragraph, I am discussing possible positive and negative biases. I am not trying to discourage anyone from installing the plates under partial or total shade. On the contrary, I stated L150- "Of course, if the plate B is placed at "symmetrical" emplacement of the plate C, the positive daily imbalance of plate B is then opposite of C plate imbalance, averaging these two plates will recover the accurate measurements. This is one of the reasons to have numerous plates installed." Based on my personal experience I am just noticing that "naturally" we tend to avoid shadowed surfaces. I am not cautioning it. That said, in the case of *obvious singular* inhomogeneity, I think that this position should be avoided. In a savanna, the trees' shadow is not a *singular* inhomogeneity.

RC1: "L200: 'the deep roots absorbed water has a lower temperature than the soil surface temperature.' Not always, in winter the contrary happens."

Absolutely, in the winter. However, during cold seasons, vegetation water use is much lower than it is in the hotter seasons. I calculated the amount of water used by winter wheat during one year, and winter wheat water usage during the cold seasons is relatively small. However, I nuanced this sentence. Please note that this calculation does not imply any relative temperature assumptions.

RC1: "L231: Rainfall has always a negative effect or it could be positive, for instance when the rain is liquid and the soil is frozen?"

Of course, this is a possibility. I did not record any corresponding occurrence on FR-Lam but this is a possibility. Please note again that for corresponding calculations no assumption on relative temperatures (rain temperature versus soil surface temperature) is needed.

RC1: "L270: I cannot understand why the geothermal flux should be added or removed from the measured flux."

I agree that this point was not clearly described and can be developed in the revised version. If for the SEB equation we are using overall SHFP measured G flux with subtracted geothermal flux $G^c = G - G_{Th}$, then we have to add $G_{Th}$ to the SEB equation (please note that $G_{Th}$ is negative). If it was subtracted from SHFP measured thermal flux only for missing convective soil heat flux assessing purpose but in the SEB equations we are using the raw SHFP measurement G then $G_{Th}$ do not have to be added to the SEB equation.

RC1: "Figure 10: Could you place more intuitive units along the X-axis? 2.5 *10^7 s is about 289 days."

I have changed the units from seconds to days. The reason to use seconds rather than days or any other units is that a linear fit of J/m² versus seconds provides directly a slope in w/m².

RC1: "L293: I am getting lost here. I do not see any graph, Figure 6 was earlier, depicting the rainfall effect."

An error in the figure number in the text. It should be mentioned "10", not "6".

RC1: "L306: 'It is important to carefully chose the installation place and check the possible imbalance by a yearly integration.' Besides 'chose' (choose), I disagree with the concept expressed. I would prefer a fully random selection of the location places or stratified sampling, but always avoiding subjective 'expert selection'."

Effectively my sentences can be misinterpreted. Rather than privileging "emplacement designations" I am privileging "emplacement exclusions" such as near the pit for soil water content probes or instrument enclosures. These pits or enclosures are not natural inhomogeneities but results from our (scientific) activities, are invasive and do not represent natural inhomogeneities existing on the studied plot. Random placement or other protocols are not discussed here and are beyond the scope of this paper. I would suggest "choosing" the way to place the plates (randomly or another way) at *natural* locations and *check* "a posteriori" if it was a "judicious choice" as even randomly placed plates may not equally represent heat fluxes affected by the inhomogeneities boundaries or be biased by SHFP malfunctions (SHFP cables can be deteriorated by animals causing intermittent malfunctions and so on).

RC1: "L327: The possible role of water table temperature is not discussed in the text."

Water table temperature was used to assess the $G^c$ imbalance caused by evapotranspiration (l214-).

I would like to answer, point by point, the second referee's RC2 comments.

RC2: "In section 3.2.3 Evapotranspiration, positive imbalances sources, I would recommend the author to include some tought on hydraulic redistribution (i.e., hydraulic lift). This might be a good source of discussion, and some comments might be worthed https://doi.org/10.1016/S0169-5347(98)01328-7"

The hydraulic lift is effectively an interesting mechanism of groundwater transfer between the deep wet soil layers and shallow dry soil layers. The suggested reference is added to the revised manuscript. Indeed, depending on how shallow is the soil layer benefiting from the hydraulic lift, namely if the root lifted water is released below or above the SHFP, the corresponding convective heat flux should be added or not to the sensed heat flux.

RC: "In section 3.2.4 Rainfall or irrigation a negative and positive imbalance source, it will be good to include a perspective of non-rainfall/irrigation inputs (i.e., mist, fog, marine brezee). It is a common feature of some mediterranean ecosystems (i.e., across the shoreline of the Californias), and might enhance the audience. The mist/fog/marine brezee is an important input of water that is not traditionally measured as an input, but is measured as an output of energy by eddy covariance measurements, however, might have influence in the soil heat flux too."

Anty convective flow means any fluid movement into the soil carrying some energy, "escapes" from SHFPs' measurement. In this paper, I would highlight the measurement leaks resulting from that fluxes citing the most obvious and present on FR-Lam. It is beyond the scope of this paper to make an exhaustive list of the possible convective flux. However, I agree with the RC2 that it may initiate a larger discussion signaling some possible convective fluxes even if they are not studied in this paper. A corresponding sentence and references are added to the conclusion in the revised version of the manuscript. As I have already signaled in my answer to RC1, snowfall and hail fall should be considered too, I agree that mist/fog and marine breeze should be considered for

concerned climates. As I am saying in the text, all these considerations may deserve more attention.

RC2: "L185-190. I do not completely agree on the criteria for removing G42 and G51. From my point of view, this is natural variability until it is demonstrated that the sensors are incorrect. Is there another criteria for removing these measurements? i.e., do not fall between three standard deviations from the mean of the remaining sensors?"

This point is similar to the reticence expressed by the first referee. I am not dealing with any malfunctioning SHFP here. The main reason for the proposed exclusion is the non-representability of the plates G42 and G51 which display a very positive imbalance when the close placed plates (the plates are always placed by pair and spaced by 60cm on Fr-Lam) display a common imbalance. Furthermore, in my mind, G42 and G51 are not resulting from a correct measurement of a non-representative emplacement, but a not correct measurement of a non-representative emplacement. Indeed, the SHFP are measuring only the vertical (one-dimensional) conductive heat flow. In the case of one-dimensional conductive heat flow, after geothermal heat flow subtraction, the annual integration should be almost nil. This is never the case and the main two causes are:

- Presence of horizontal heat fluxes resulting mainly from a narrow soil or energy apport inhomogeneity such as a partially shadowed surface.
- Convective, not sensed, heat fluxes such as root pumped water, rainfall water infiltration, and so on.

Individually integrating all the SHFPs' measurements and comparing the results provide a rapid indication of the inhomogeneities presence in the case of obvious divergence of several SHFPs' measurements compared with most other plates' measurements (G42 and G51 in this case). These inhomogeneities can be real and natural, it is not the point, however, the measurement of these inhomogeneities maybe not correct with the SHFPs because the non-vertical heat fluxes are not sensed. Resulting SHFPs' measurements are partial. Moreover, any inhomogeneity causes on one side of the boundary a positive unbalance and a negative unbalance on the other side. When an SHFP is located on one side without another SHFP on the other side, the overall measurement does not represent the overall soil heat flux but only the very local heat flux. The only way for correct inhomogeneity measurement would be to place at least two SHFPs on both sides of the boundary providing a correct *overall* measurement. But this has to be checked with annual integration.

In the case when SHFP were placed only on one side I am proposing to reject the corresponding measurements to assess the other measurement imbalance source: the convective heat fluxes. This assessment is, in my mind, possible when most of the plates present a similar yearly imbalance as in the studied FR-Lam example.

---

## Author Response (AR2)

Dear associate Editor, Dear Editorial Team,

I am submitting a modified version of my paper about the soil heat fluxes measurement with the soil heat fluxes plates (SHFPs) and energies that should be included in the surface energies balance (SEB) equations. Since I can't change the fundamental law of energy conservation, I did not retrieve the concerned result: annual integration nullity of the surface energy exchange after geothermal flux subtraction. This point results from a very simple calculation of an inert body energy conservation. However, I did develop the corresponding calculation explanation in Appendix A to allow all readers to follow, step by step, this calculation. In another appendix, Appendix B, I am recalling the different component so f the heat exchanges: conductive, radiative, and convective. Usually, the convective component is neglected in the soil and I am describing a simple case of advective heat flux into the air where the convective exchanges are predominant. In the soil, the convective part of the heat flux is not predominant but neither negligible.

Both appendixes are destinated to clarify the fact that there is always a difference between the reality governed by the physical laws and the measurement governed by the technical limitations. The energy conservation law is describing the real heat exchanges when the SHFPs are measuring only one heat fluxes exchanges component: conductive fluxes, therefore, limiting its real fluxes representation accuracy.

Honestly, I think that I have already answered my referee's questions and objections and developed these points in the first revised text. My feeling is that Dr. Montagnani dos not even read the revised text nor accords much attention to my answers but I would like to recall here why I am persuaded that there is a big misunderstanding and answer his last comments point-by-point.

After I queried clarifications of the reasons driving my first referee Dr. Montagnani, named hereafter RC1, to recommend the rejection of my paper, I obtained the following objections:

RC1: "Where for me the author makes a serious mistake is precisely when he claims that the balance is zero at the annual level. It contradicts centuries of experimental physics, and also world maps of geothermal energy fluxes."

Annual integration nullity of a geothermally corrected soil surface heat exchange is not a hypothesis that I make but a result of a simple calculation based on a fundamental physical law provided in Appendix A, already added to in the first revised version of my paper. Denying this result amounts to denying the fundamental law of energy conservation. RC1 is justifying the rejection of these results by the "contradiction with centuries of experimental physics". Besides that the soil heat exchange measurements are relatively recent; it seems difficult to sustain a fundamental physical law inexactitude due to a partial experimental measurements mismatch. The convective soil heat fluxes such as vegetation transpiration causing ground-water moves are real and not sensed by the SHFP nor any other known sensor. Corresponding energy losses are not negligible. Consequently, the annual integration of the SHFPs' measurements, after geothermal flux subtraction is not nil and the difference gives us the importance of the unmeasured heat fluxes. Concerning a "contradiction with world maps of geothermal energy fluxes", I am sorry, I do not understand what RC1 is talking about. The estimation of the geothermal energy at our station Fr-Lam (South-west of France) was taken from the studies of SIG BRGM realized in France in 1989 and widely used for geothermal fluxes estimation. Maybe the objection of CR1 is coming from the sign "-" that I attribute to this flux when

the dedicated measures are mainly talking about positive fluxes. The sign change is due to the convention adopted where an upgoing flux is given as negative.

CR1: "Instead of excuting some heat flux plates and forcing the balance to zero, I would do an error analysis with Monte Carlo sampling and try to determine based on the error that is deemed acceptable how many heat flux plates to put in. That way you circumscribe the random error to a defined value."

There is always this misunderstanding that is misleading RC1. I do not force anything. The balance of the real surface heat flux, after geothermal subtraction, is nil (this is according to the fundamental energy conservation law). The balance of the SHFPs' is not nil because it cannot be because the SHFP is not sensing all the fluxes. If I am proposing to exclude some plate's measurements it is not "to force the balance to zero", no one plate's balance is zero, but I am proposing to exclude some plates measurements if their measurements are very different from the mean measurement, only to exclude the inhomogeneities perturbations (could be seen as random error) and to assess the unmeasured heat fluxes (could be seen as systematic error). I have already signaled the difficulties to use statistics on a very limited number of plates. We cannot ignore the feasibility of the plate's multiplication. All the corresponding explanations were already developed in the first revised version of my paper and the answers to the RC1's first rapport.

RC1: "Also, to reduce any bias from random placement, you could do a measurement period with many plates, then reduce them in number after stratifying them (stratified sampling). But always accepting the experimental results, only possibly removing outliers. If, on the other hand, there is a systematic error, it should be pointed out and characterized as such. Systematic error is not reduced by increasing the sampling points."

It is exactly what I am answering to the first RC1's comments and adding explicitly to the first revised version. There are two different sources of SHFP's measurements imbalance: spatially random perturbations caused by inhomogeneities and rather spatially homogeneous perturbations caused by convective fluxes. Forming a mean measured flux and then discarding the plates with measurements being far from the mean measured flux is to approach the systematic perturbation which is the convective unmeasured fluxes. Concerning the sentence: "but always accepting the experimental results, only possibly removing outliers" this is not at all the usual procedure when an overall representative measurement is sought after but a punctual perturbation is detected. For example, the eddy covariance measurements are often discarded not because they are outliers but just because we know that the conditions were not met for eddy covariance technique optimization such as too stable conditions, etc. Eddy covariance measurements are discarded to not bias the accumulated measurements. Disrupted measurements are then fulfilled by gap-filling methods where measurements are "recreated" from the previously acquired measurements. We are far, very far, from "always accepted measurements". Also, I already mentioned, in the answers and the revised text, that in the case of an inhomogeneity boundary proximity, the soil heat flux is no more vertical or the SHFPs are always placed horizontally which means that they measure the vertical component of the heat flux. If the heat flux is not vertical SHFP's measurement is biased. Should we accept all the measurements even if we know that they are biased? An extensive discussion about the inhomogeneities and their boundaries influence is already in the answers to the RC1's first report and the first revised text.